# Trajectory Improvement and Reward Learning from Comparative Language Feedback

**Zhaojing Yang**[1]  **Miru Jun**[1]  **Jeremy Tien**[2]
**Stuart J. Russell**[2]  **Anca Dragan**[2]  **Erdem Bıyık**[1]

[1]University of Southern California  [2]University of California, Berkeley

**Abstract:** Learning from human feedback has gained traction in fields like robotics and natural language processing in recent years. While prior works mostly rely on human feedback in the form of comparisons, language is a preferable modality that provides more informative insights into user preferences. In this work, we aim to incorporate comparative language feedback to iteratively improve robot trajectories and to learn reward functions that encode human preferences. To achieve this goal, we learn a shared latent space that integrates trajectory data and language feedback, and subsequently leverage the learned latent space to improve trajectories and learn human preferences. To the best of our knowledge, we are the first to incorporate comparative language feedback into reward learning. Our simulation experiments demonstrate the effectiveness of the learned latent space and the success of our learning algorithms. We also conduct human subject studies that show our reward learning algorithm achieves a $23.9\%$ higher subjective score on average and is $11.3\%$ more time-efficient compared to preference-based reward learning, underscoring the superior performance of our method. Our website is at https://liralab.usc.edu/comparative-language-feedback/.

**Keywords:** Learning from human feedback, reward learning, HRI

## 1 Introduction

Learning from human feedback has gained significant popularity in robotics, leading to the study of different forms of human feedback: demonstrations [1, 2, 3, 4], preference comparisons [5, 6, 7, 8], rankings [9], physical corrections [10], visual saliency maps [11], human language [12, 13], etc. Among these, preference comparisons grew popular for its simplicity and ease of use, especially compared to demonstrations [7]. Preference comparisons often involve users choosing between a pair of choices. Using these selections to learn a reward function and train a policy is known as reinforcement learning from human feedback (RLHF) [6] or more generally preference-based learning [5, 14]. It has proven applicable to a broad range of fields ranging from robotics [7, 15] to natural language processing [16], from traffic routing [17] to human-computer interaction [18].

Despite their successes, preference comparisons suffer from problems [19] such as the unreliability of human data and the limited information bandwidth, i.e., each pairwise comparison contains at most 1 bit of information [20]. There has been research to provide a better interface [21], allowing the users to specify their preferences for every feature, but they require features to be designed by hand. As an alternative form of human feedback, comparative language is considerably more informative than preference comparisons, allowing users to prioritize specific aspects. For example, it allows users to naturally indicate their preference about speed by simply saying, "the robot should move faster," making it more intuitive and interpretable.

In this work, we aim to leverage comparative language feedback to learn the human preferences, i.e., their reward functions. In pursuit of this objective, we first learn a shared latent space that aligns trajectories and language feedback. This alignment enables the robot to comprehend human language feedback, leverage it for adapting its behavior to learn and better align with the human's preferences. To test the effectiveness of our approach, we conduct experiments in two simulation environments and a human-subject study with a real robot. The results suggest that reward learning from

8th Conference on Robot Learning (CoRL 2024), Munich, Germany.

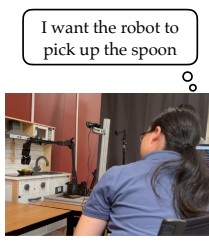
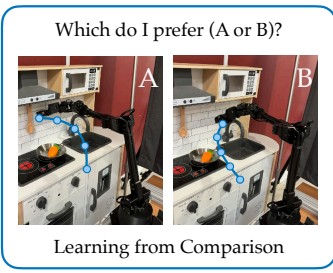
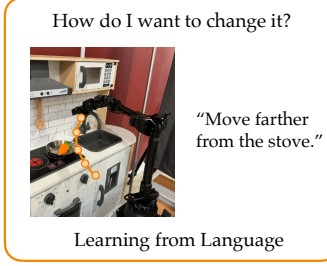

Figure 1: An image from our human subject studies where the human user wants the robot to pick up the spoon. Compared to traditional comparison preference learning, our language preference learning enables users to give more informative feedback, which helps the robot to capture human preferences more efficiently.

comparative language feedback outperforms traditional preference comparisons in performance and time-efficiency, and is favored by most of the users.

## 2 Related Work

Before formally defining the problem, we will first review existing works in learning from human feedback, preference-based learning, and robot learning from human language feedback.

**Learning from Human Feedback.** Several promising approaches leverage human feedback to train robots [2, 3, 4, 5, 6, 7, 8, 9, 10, 11, 12, 13, 22]. However, the most popular methods of learning from demonstrations and rankings involve a trade-off between informativeness and ease of use [6, 19, 23, 24]. Furthermore, none has taken advantage of the expressive nature of language to develop a method of reward learning from comparative language feedback. Addressing this gap will provide more scalable methods for training robots, particularly in complex environments where traditional feedback forms may be insufficient or impractical.

**Preference-based Learning.** Based on preference comparison feedback, preference-based learning is widely used for its logical intuitiveness and ease of use. Learning from rankings is beneficial to deep reinforcement learning [6], but suffers from several shortcomings [19] such as the reliability of human data and difficulties of choosing from equivalent choices. Works have sought to improve its shortcomings in time- and sample-efficiency by actively generating pairs based on information gain [20] or volume removal [5], and doing these in batches of pairs [25]. Sikchi et al. [26] and Brown et al. [27] integrate preference feedback into imitation learning, demonstrating superior performance in their respective approaches. However, the fundamental limitation remains that each pairwise comparison provides at most 1 bit of information [20]. In addition, users struggle to choose between two similar trajectories [7]. In contrast, our method of comparative language feedback offers more informative input after observing just one trajectory, while maintaining intuitiveness and ease of use.

**Robot Learning with Human Language Feedback.** Benefiting from advancements in natural language processing, works have leveraged language for adjusting robot trajectories [13, 28, 29, 30, 31, 32], fine-tuning language models [12], and reward shaping [33, 34]. For example, Shi et al. [31] use language-conditioned behavior cloning (LCBC) for corrective language commands and improving policies. Lynch et al. [35] describe an approach for real-time guidance from humans using natural language in order to achieve a goal. Cui et al. [36] introduce an approach to use human language feedback to correct robot manipulation in real-time via shared autonomy. Goyal et al. [33] utilize human language combined with past action sequences to generate rewards, and in a separate work, Goyal et al. [34] leverage natural language to map image observations to rewards. However, these works focus on using human language as the instruction or correction to the robot, none of them has explored learning the reward function of humans entirely from comparative language feedback, which offers benefits over policy learning with generalizability and explainability. Additionally, our method iteratively updates the learned reward function for better preference alignment, providing an advantage over one-shot corrections [13, 29].

## 3 Problem Definition

Having reviewed the related literature, we are now ready to formally define the problem. We model the robot as a decision making agent in a standard finite-horizon Markov decision pro-

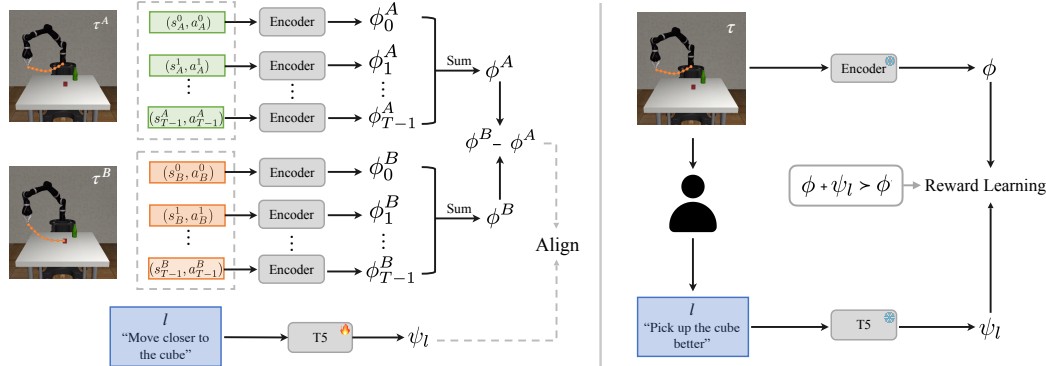

|  (a) Model Architecture  |  (b) Language-based Reward Learning  |

Figure 2: Overview of our approach. (a) Architecture of the model that learns a shared latent space between trajectories and comparative language feedback. (b) Comparative language-based reward learning.

cess (MDP). Each robot trajectory consists of state-action pairs for $T$ time steps[1]: $\tau = \{(s_0, a_0), (s_1, a_1), ..., (s_{T-1}, a_{T-1})\}$. A reward function $R(s_t, a_t)$, which is only known by the human user, encodes human preferences regarding the task. The reward of a trajectory is defined as: $R(\tau) = \sum_{t=0}^{T-1} R(s_t, a_t)$. For each trajectory, the human may provide language feedback $l$ that attempts to improve the trajectory (based on the reward function) in one aspect, e.g., speed, distance to objects, height of the robot's arm, etc. As an example, the human might say "move farther from the stove" (see Figure 1) if the robot's trajectory would be higher-reward in that way. Our goal is to develop a framework where we learn the reward function based on such feedback.

Unfortunately, this task is doomed without any information about how language feedback relates to the different aspects of the task. In this work, we learn this relation with an offline pretraining phase where we collect a dataset that consists of pairs of robot trajectories and a language label describing how the two trajectories differ. We use this dataset to learn encoders that map trajectories and language feedback onto a shared latent space. We leverage these encoders to learn the preferences (reward functions) of different users based on their language feedback.

In the next section, we detail our approach to learning the latent space and explain how this learned latent space is utilized for iterative trajectory improvements and reward learning.

## 4 Our Approach

As shown in Figure 2, our approach is composed of two stages: First, we learn a shared latent space where robot trajectories and human language feedback are aligned. Second, we leverage this learned latent space to (1) improve robot trajectory or (2) learn human preferences.

### 4.1 Learning the Shared Latent Space

To learn a shared latent space for trajectories and language feedback, we collect a dataset of $(\tau^A, \tau^B, l)$ tuples, where $\tau^A$ and $\tau^B$ are a pair of trajectories and $l$ is a language utterance that describes the difference between the two trajectories. Note that this language utterance does not necessarily align with the human's preferences about the robot: it just describes a difference between the trajectories — it is possible to have $l =$"move faster" if the robot moves faster in $\tau^B$ than $\tau^A$ even though users want the robot to move slowly.

After collecting such a dataset, we propose the model visualized in Figure 2 to learn the shared latent space between trajectories and language utterances. Similar to [37], we use a neural network to encode each state-action pair $(s_t, a_t)$ from the pair of trajectories, $\tau^A$ and $\tau^B$, to embeddings $\phi_t^A$ and $\phi_t^B$, respectively. We want these embeddings to contain information about aspects of the trajectories that the human may care and give feedback about.

To achieve this, we align the difference between the embeddings $\phi^A$ and $\phi^B$ with embeddings $\psi_l$ of the language feedback $l$ by using the following loss function:

$$L_{\text{align}}(\tau^A, \tau^B, l) = -\log\left(\text{sigmoid}\left(\psi_l^\top(\phi^B - \phi^A)\right)\right) \qquad (1)$$

---

[1]Our work trivially extends to the cases where trajectory length is not fixed.

where the sigmoid can be considered as the probability that the language utterance $l$ is inputted for the pair of trajectories $\tau^A$ and $\tau^B$. Intuitively, when $\phi^B - \phi^A$ and $\psi_l$ align, i.e., have the same direction for fixed magnitudes, the loss is minimized. This alignment will enable us to acquire the embedding of an improved trajectory as $\phi^A + \psi_l$ when given an initial trajectory $\tau^A$ and comparative language feedback $l$.

Instead of training a language encoder from scratch, which would require a prohibitively large and diverse dataset of $(\tau^A, \tau^B, l)$ tuples, we use a pretrained T5 model [38]. To align embeddings of trajectories with those of language feedback, we first freeze T5 model and train the trajectory encoder. Subsequently, we perform co-finetuning of both components.

Additionally, we incorporate a normalization term into the loss function to help balance the magnitude of the embeddings. This term constrains the norms of the trajectory embeddings to remain below 1 and the norm of the language embeddings to be close to 1:

$$L_{\text{norm}}(\tau^A, \tau^B, l) = a \cdot \left( \max\{\|\phi^A\| - 1, 0\} + \max\{\|\phi^B\| - 1, 0\} \right) + b \cdot \|\psi_l - 1\|^2 \qquad (2)$$

where $a$ and $b$ are two hyperparameters.

Overall, the objective we use in the model training consists of two terms:

$$L(\tau^A, \tau^B, l) = L_{\text{align}}(\tau^A, \tau^B, l) + L_{\text{norm}}(\tau^A, \tau^B, l), \qquad (3)$$

which we use to train the trajectory encoder and finetune T5. Training this architecture gives us the ability to encode any trajectory and language utterance in the same latent space. In the next subsection, we demonstrate how this latent space is useful for iteratively improving robot trajectories and for learning human preferences based on their language feedback.

### 4.2 Utilizing the Learned Latent Space

The shared latent space aligns robot trajectories with human language feedback. This alignment enables an intuitive understanding of user preferences. We will now explore two primary ways to leverage this learned latent space: first, to iteratively improve the robot's trajectory, and second, to accurately learn user preferences.

#### 4.2.1 Iterative Trajectory Improvement

Firstly, we leverage the latent space to iteratively improve an initial suboptimal robot trajectory. We start with showing the initial trajectory $\tau^0$ to the user and asking for language feedback $l^0$. Upon receiving human's language feedback, we use our trained encoders to compute the trajectory embedding $\phi^0$ and language embedding $\psi_{l^0}$. We then find the *improved trajectory* $\tau^1$ such that its difference with $\tau^0$ best aligns with the human's language feedback based on cosine similarity:

$$\tau^1 = \operatorname*{argmax}_{\tau'} \frac{\psi_{l^0}^\top (\phi' - \phi^0)}{\|\phi'\|_2 \cdot \|\phi^0\|_2} \qquad (4)$$

We iteratively continue this process for $N$ iterations to obtain $\tau^0, \tau^1, \ldots, \tau^N$. In this work, we search over a discrete, predefined set of trajectories to solve the optimization in Eq. (4) for computational efficiency. It is, however, possible to use reinforcement learning or model-predictive control algorithms to solve this optimization at the expense of increased computational cost.

#### 4.2.2 Reward Learning from Comparative Language Feedback

In addition to improving trajectories, we also utilize the learned latent space to learn the user's preference, i.e., their reward function. Previous approaches ask users to label their preference from a pair of trajectories, which requires the users to watch two trajectories in order to give at most 1 bit of information. In our language-based reward learning approach, for each query $i$, we only show users one trajectory $\tau_i$ to collect language feedback $l_i$ based on their preferences. Given one trajectory, we construct an improved trajectory $\hat{\tau}_i$: $\phi_{\hat{\tau}_i} = \phi_{\tau_i} + \psi_{l_i}$. Note that $\hat{\tau}_i$ is just an imaginary trajectory that maps to $\phi_{\tau_i} + \psi_{l_i}$ in the learned latent space. Then following the Bradley-Terry model [39], which is commonly used in preference-based learning, we model a preference predictor with the reward function $r_\xi$ as:

$$P_\xi(\hat{\tau}_i \succ \tau_i) = \frac{\exp r_\xi(\phi_{\hat{\tau}_i})}{\exp r_\xi(\phi_{\hat{\tau}_i}) + \exp r_\xi(\phi_{\tau_i})} \qquad (5)$$

where $r_\xi$ is a neural network parameterized with $\xi$. The reward function $r_\xi$ is trained by minimizing the following negative loglikelihood loss:

$$L_{\texttt{Explicit}} = -\frac{1}{n} \sum_{i=0}^{n-1} \log P_\xi(\hat{\tau}_i \succ \tau_i). \tag{6}$$

We now make an important observation about the comparative language feedback. When the user tells the robot to move farther from the stove, it indeed indicates a preference between the original trajectory and the improved trajectory that moves farther from the stove. However, this feedback contains much more information than this comparison. The user could use any comparative language utterance to teach the robot, but they specifically selected one about the distance to the stove. This means, with high probability, the improvement the robot may get from this feedback is higher than that of any other comparative language feedback.

Mathematically, this indicates a preference for $\hat{\tau}_i$ over $\tilde{\tau}_i : \phi_{\tilde{\tau}_i} = \phi_{\tau_i} + \psi_{\tilde{l}_i}$ where $\tilde{l}_i$ is any language utterance other than $l_i$. Again we apply the Bradley-Terry model to utilize this implicit preference:

$$P_\xi(\hat{\tau}_i \succ \tilde{\tau}_i) = \frac{\exp r_\xi(\phi_{\hat{\tau}_i})}{\exp r_\xi(\phi_{\hat{\tau}_i}) + \exp r_\xi(\phi_{\tilde{\tau}_i})} \tag{7}$$

Based on this, we sample $k$ language feedback from a set of pre-collected language utterances other than $l_i$, and minimize the following loss about the chosen language feedback:

$$L_{\texttt{Implicit}} = -\frac{1}{k} \sum_{j=0}^{k-1} \log P_\xi(\hat{\tau}_i \succ \tilde{\tau}_{i,j}) \tag{8}$$

Overall, the loss for reward learning from comparative language feedback is as follows:

$$L_{\texttt{Reward}} = L_{\texttt{Explicit}} + L_{\texttt{Implicit}} = -\frac{1}{n} \sum_{i=0}^{n-1} \Big( \log P_\xi(\hat{\tau}_i \succ \tau_i) + \frac{1}{k} \sum_{j=0}^{k-1} \log P_\xi(\hat{\tau}_i \succ \tilde{\tau}_{i,j}) \Big). \tag{9}$$

Training the reward function with this loss enables us to efficiently capture human preferences through comparative language feedback.

## 5 Experiments

We conducted simulation experiments and human subject studies to validate our methods with diverse tasks and features that humans may give feedback about.

### 5.1 Simulation Experiments

**Synthetic humans.** For simulation experiments, we synthesize the comparative language feedback from a simulated human with a simplified reward function $R$:

$$R(s_t, a_t) = w^\top \theta(s_t, a_t) \tag{10}$$

where $\theta$ is a function that maps a state-action pair to a vector of high-level features (e.g., speed, distance to the stove). We similarly define $\theta(\tau)$ to denote the sum of features over a trajectory $\tau$. $w$ is a vector of weights that maps the features to a scalar reward value. Both $\theta$ and $w$ are unknown to our algorithm.

Given a true reward function $w$, let $\tau^*$ be the optimal trajectory under reward $w$. Then, the synthetic human gives the noisy language feedback $l^0$ when shown trajectory $\tau^0$:

$$l^0 \sim \ell \left( \text{softmax}(w \odot (\theta_l^* - \theta_l^0)) \right) \tag{11}$$

where $\theta^*$ and $\theta^0$ denote the true cumulative features of $\tau^*$ and $\tau^0$ respectively, $\odot$ denotes element-wise multiplication. The function $\ell$ takes a sample from the output of the softmax, which is computed across different features

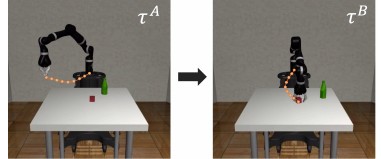

*"Move closer to the cube"*

Figure 3: Each dataset sample is a pair of trajectories and a language feedback.

we designed for the simulated humans (see Appendix A), and outputs a language feedback that corresponds to the sampled feature. The language feedback to output is chosen randomly from all language utterances of that feature in the GPT-augmented dataset (see Appendix E).

**Environments.** We experimented in two simulation environments: Robosuite [40] and Meta-World [41]. Robosuite has a Jaco robot arm at a table with a cube and a bottle, and the task is to pick up the cube. The states are given as $640 \times 480$ RGB images (resized to $224 \times 224$) and 25-dimensional

proprioception. The action space is 4-dimensional (the end-effector always points down). Meta-World has a Sawyer robot arm at a table, and the task is to push a button on the side of the table. The states are given as $480 \times 320$ RGB images (resized to $224 \times 224$) and 20-dimensional proprioception. The action space is 4-dimensional (the end-effector always points down).

**Learning the Latent Space.** Lists of language feedback were created for hand-crafted features (see Appendix A) to indicate a change (e.g., {'Move higher', 'Move lower'} for height). They were augmented with GPT 3.5 [42] to 629 sentences for Robosuite and 660 for Meta-World. The splits are 480, 74, 75 and 492, 84, 84 for the training, validation, and test sets. Note these features are only for synthetic dataset creation — training our architecture does not require hand-designed features.

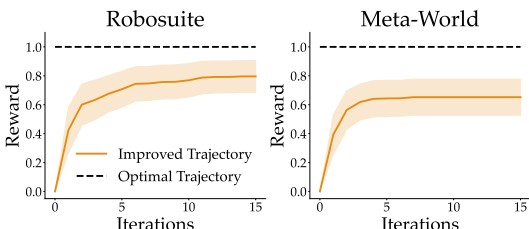

Figure 4: Results of experiments where we use simulated human language feedback to iteratively improve a robot trajectory (mean $\pm$ std over 100 runs). The dashed line represents *average* reward of optimal trajectories.

We trained RL policies with randomized weights $w$ using stratified sampling [43] over the features, then generated rollout trajectories with timesteps $T = 500$ for each environment. For Robosuite, we generated 448 unique rollouts and 324 for Meta-World. The splits were 359, 44, 45 and 260, 32, 32 for training, validation, and test sets. Trajectories were paired within each set and matched with a language feedback that describes the change in each feature from $\tau^A$ to $\tau^B$ (see Figure 3).

We train the encoders with the loss in Eq. (3), and use accuracy as the metric to evaluate:

$$\text{Acc} = \frac{|\{(\tau^A, \tau^B, l) \mid \psi_l^\top (\phi^B - \phi^A) > 0\}|}{\text{Total number of samples}} . \tag{12}$$

Training the encoders with the training set of trajectories and language utterances, we observed a test accuracy of $84.9\%$ in Robosuite and $82.9\%$ in Meta-World. This indicates the trajectory and language embeddings are well-aligned. We found co-finetuning both trajectory encoder and language model, versus utilizing a frozen language model and only training the trajectory encoder, helps map trajectories and language utterances to the same latent space (see Table 2 in the Appendix).

**Iterative Trajectory Improvement.** Next, we conducted iterative trajectory improvement experiments based on the method we presented in Section 4.2.1. We set the number of iteration steps $N = 15$ and repeat the full experiment over 100 random seeds. The true rewards of these trajectories are shown in Fig. 4. In both environments, we consistently improve the trajectories, which showcases the effectiveness of the learned latent space and the improvement algorithm. However, it can be noted that our algorithm cannot reach the performance of the optimal trajectory. This is because every improvement iteration is completely independent from the previous iterations and the robot may get stuck in a loop between good, but non-optimal, trajectories.

Our reward learning algorithm presented in Section 4.2.2, on the other hand, utilizes the entire history of human feedback, so it should not suffer from this problem. We will now demonstrate this.

**Reward Learning.** We again randomly initialize the reward weights $w$ to simulate human feedback. For each environment, we simulate three synthetic humans, and run the experiments with three random seeds for each simulated human. We use the loss in Eq. (8) to learn the reward function. We adopt cross-entropy $CE(P_w, P_\xi)$ as the evaluation metric, where

$$P_w(\tau^A \succ \tau^B) = \frac{\exp w^\top \theta(\tau^A)}{\exp w^\top \theta(\tau^A) + \exp w^\top \theta(\tau^B)} , \; P_\xi(\tau^A \succ \tau^B) = \frac{\exp r_\xi(\phi^A)}{\exp r_\xi(\phi^A) + \exp r_\xi(\phi^B)} \tag{13}$$

Another metric is the true reward value of the optimal trajectory selected from the test set based on the learned reward, reflecting how close the learned reward is to the true reward. All reward values are normalized between 0 and 1.

We compare our method against reward learning from comparisons, where for each query, the simulated user chooses a preferred trajectory from pair $(\tau^A, \tau^B)$ and with probability shown in Eq. (13).

As indicated in Figure 5a, the cross-entropy of our method decreases faster than that of comparison preference learning in both environments, demonstrating that our approach converges more quickly than the baseline. Meanwhile, Figure 5b shows that the true reward of optimal trajectories reaches

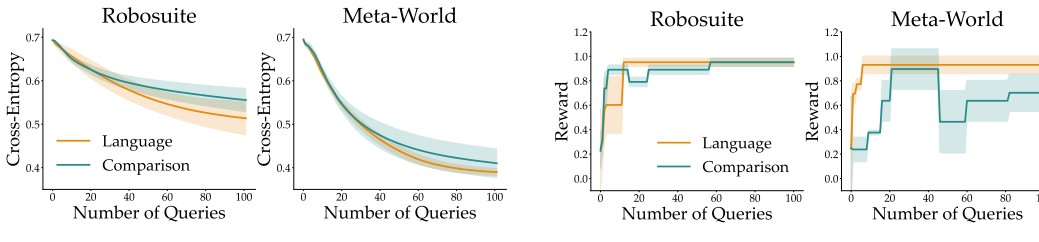

(a) Cross-Entropy        (b) True Reward Values of Optimal Trajectory

Figure 5: Results of reward learning, averaged over 3 seeds. (a) Cross Entropy: our method converges faster. (b) True Reward of Optimal Trajectory: Our approach.

a value of 1 in a shorter time with our language-based method, especially in Meta-World, further supporting that it learns the reward function more efficiently.

**Ablation Study.** We conducted an ablation study to evaluate the different components of the loss function (Eq. (8)). As indicated in Figure 6a, the combination of both components outperforms using each component individually, demonstrating the validity of our loss function design. Also, using both components fits the true reward function better, as shown in Figure 6b.

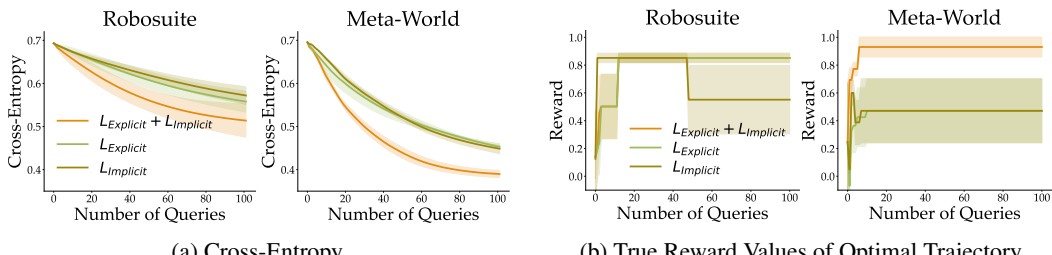

(a) Cross-Entropy        (b) True Reward Values of Optimal Trajectory

Figure 6: Results of Ablation Study. (a) Baseline Comparison: our method consistently outperforms the baseline. (b) Ablation Study: Using two components is better than applying each of them alone.

## 5.2 User Studies

To verify the effectiveness of our approach, we conducted human subject studies by recruiting 10 subjects (4 female, 6 male) from varying backgrounds and observing them to interact with our real robot setup. Our study has been approved by the IRB office of the University of Southern California.

**Real Robot Setup.** We used a WidowX 250 6DOF robot with the setup of [44] in front of a kitchen set (Figure 7). The task is to pick up a spoon while avoiding going over a pan on the stove. The states are given as $480 \times 320$ images resized to $224 \times 224$, and 22-dimensional proprioception. The actions are 7-dimensional (6DOF+gripper). To train the latent space, we collected a dataset of 321 trajectories with varying levels of success at picking up the spoon, avoiding the pan, and speed. These trajectories were divided into 192 for training, 64 for validation, and 65 for testing. 496 GPT-augmented sentences were split into 297 for training, 99 for validation, and 100 for testing. These sentences were paired with the trajectories in the same manner as in the simulation experiments. Another set of 32 trajectories was collected for the user studies following the pretraining phase.

**Study 1: Trajectory Improvement.** Users start with a suboptimal trajectory from the dataset and give language feedback for a maximum of 10 iterations until they are satisfied.

**Study 2: Preference Learning.** For 20 iterations, users give feedback (preference comparison or comparative language) on random trajectories and rate the optimal trajectory from the dataset with respect to the learned reward after every 5 iterations. To avoid bias, 5 users began with the comparison based method and 5 with the comparative language based method.

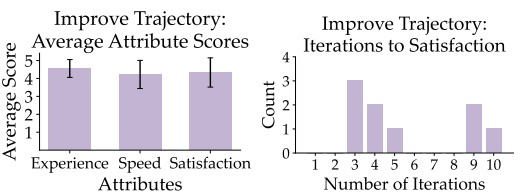

Figure 8: (Left) Average attribute scores. (Right) Iterations to satisfaction.

After each study, users were asked to complete post-study surveys (see Appendix B).

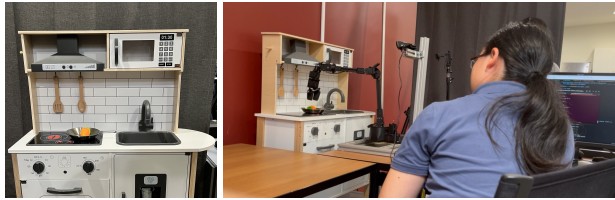

Figure 7: (Left) Closeup view of the kitchen set with the spoon hanging on the wall, above the pan on the stove. (Right) WidowX 250 6DOF robot arm, user, and text prompt for experiments.

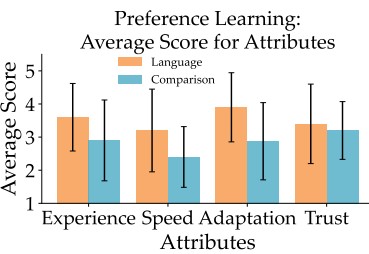

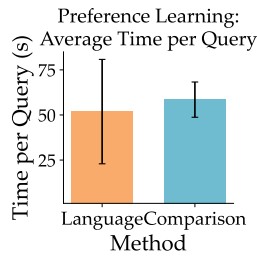

Table 1: User satisfaction in aspects, not mutually exclusive between methods

|  | Satisfactory | |
| Method | Language | Comparison |
| --- | --- | --- |
| Time Efficient | 9 / 10 | 1 / 10 |
| Convenient | 6 / 10 | 8 / 10 |
| Adaptable | 8 / 10 | 3 / 10 |

Figure 9: (Left) User ratings. (Right) Time per query.

**Results.** Our post-study surveys reveal that the first study has consistently positive responses for user experience and speed of adaptation, but has two peaks in iterations to satisfaction (Fig. 8). We conjecture that the dataset of 32 trajectories may not have contained trajectories that the users desired. As for the second study, our language-based method scored better than the comparison-based method **for all attributes**. The average score of the language-based method over all attributes is **23.9%** higher than the comparison-based method , illustrating the superior performance of our approach. In terms of time-efficiency, our method takes **11.3%** less time for users to answer each query as shown in Fig. 9, even though it includes the time it took for users to type their feedback. Indeed, users found the language method to be more time efficient and adaptable but found comparison method to be more convenient (Table 1). Fig. 10 shows the user rating of optimal trajectories given by the learned reward function. To quantitatively assess the learning efficiency, we follow prior work in active learning [9, 45, 46] by checking the area under curve (AUC) for both lines. We found that the AUC for the comparative language line is statistically significantly higher than that of preference comparison line ($p < 0.05$). This indicates that our language-based approach more efficiently captures human preferences compared to the comparison-based method.

An unexpected observation from Fig. 10 is that humans' ratings decreased from 5 queries to 10 queries while using comparative language feedback. This may be because of the relatively small subject size: it is possible that the average rating after 5 iterations, where the standard deviation is indeed large, is higher than it should be. Another potential explanation is that humans may be more lenient to the robot after 5 iterations than 10 iterations, especially in comparative language feedback where they only watch the robot 5 times to give their feedback as opposed to 10 times in preference comparisons.

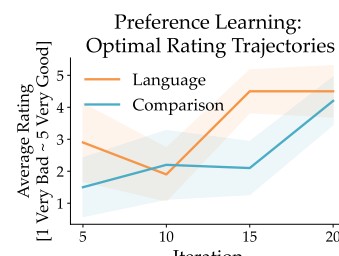

Figure 10: Average ratings of the optimal trajectory shown every 5 iterations for each method.

## 6  Conclusion

In this paper, we presented a robot learning framework to learn human preferences using comparative language feedback. For this, we aligned trajectories with language feedback in a shared latent space, then used it to improve trajectories and learn preferences. Experiments in simulation environments and real-world user studies suggest that our approach consistently outperforms comparison preference learning and is favored by most users for aspects such as time efficiency and adaptability.

**Limitations and Future Work.** Even though our user studies show some generalizability in the comparative language feedback, our model is limited by the feedback about the objects seen in the pretraining data. Additionally, we sample queries uniformly at random during reward learning. Based on these limitations, the future works include: (1) use image annotations to generate language feedback containing all objects, and (2) implement active learning to reduce the number of iterations required to reach the optimal trajectory.

**Acknowledgments**

This work was partially supported by NSF HCC grant #2310757 and a gift in support of the Center for Human-Compatible AI at UC Berkeley from the Open Philanthropy Foundation.

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

# A    Dataset Features

**Robosuite Features**

- The height of the robot's end-effector
- The speed of the end-effector
- The distance between the end-effector and the bottle
- The distance between the end-effector and the cube
- How well the robot lifts the cube (i.e., level of success of cube-lifting task)

The level of success of the cube-lifting task is quantified by: 1) whether or not the cube is lifted above the height of a success threshold or 2) a weighted sum of the distance to the cube and whether or not the end-effector is grasping the cube (the level of success in picking up the spoon).

**Meta-World Features**

- The height of the robot's end-effector.
- The velocity of the robot's end-effector.
- The distance between the end-effector and the button.

**WidowX Features**

- The velocity of the end-effector
- The distance between the end-effector and the pan
- How well the robot picks up the spoon  (i.e. level of success)

The level of success is quantified by: 1) whether the spoon is grasped from the hook 2) a weighted sum of the distance to the spoon and whether the end-effector is grasping the spoon.

# B    User Study Surveys

Here are the questions in our user study surveys.

**Pre-Study Survey.** To assess the subjects' experience and level of skill with robotics and machine learning, we conducted a short pre-study survey before any of the experiments.

1. "Age"
2. "Gender"
3. "Race"
4. "Highest level of education"
5. "Please describe your level of robotics experience, on a scale from 1 to 5."
6. "Please describe your level of machine learning experience, on a scale from 1 to 5."
7. "Have you ever interacted with a robot before? Please describe:"

**Post-Study Survey — Improve Trajectory.** Users filled out this post-study survey after completing the Improve Trajectory experiment.

1. "Are you satisfied with the final trajectory?" [1 Completely unsatisfied - 5 Completely satisfied]
2. "How many iterations did it take for the robot to adapt to your feedback?"
3. "How fast does the robot adapt to your feedback?" [1 Very slow - 5 Very fast]
4. "How did the robot's learning capabilities affect your interaction experience?" [1 Negatively - 5 Positively]
5. "Overall, do you think the approach is effective?" [Y/N]
6. "Do you have any other comments?"

**Post-Study Survey — Preference Learning.** Users filled out this post-study survey after completing two methods of the preference learning experiment.

1. "What did you like about these methods? For each aspect below, please click the circle if you found the method satisfactory in that regard:" [Adaptability, Time efficiency, Convenience, None of them]
2. "Could you explain the reasons?"
3. "How long did it take for the robot to adapt to your feedback?" [1 Very slowly - 5 Very quickly]
4. "How did the robot's learning capabilities affect your interaction experience?" [1 Negatively - 5 Positively]
5. "Did you notice the robot learning or adapting to your behavior?" [1 Not at all - 5 Constantly]
6. "How did the robot's learning process affect your level of trust in its capabilities?" [1 Negatively - 5 Positively]
7. "Overall, which method do you prefer?" [Pair-wise comparison/Language feedback]
8. "Why do you prefer this method?"
9. "Is there anything you wish the system would do but currently does not?"
10. "Do you have any other comments?"

## C  Experiments — Additional Figures

### C.1  The Effect of Co-finetuning

Co-finetuning both trajectory encoder and language model, versus utilizing a frozen language model and only training the trajectory encoder, helps map trajectories and language utterances to the same latent space. This is shown in Table 2.

Table 2: Test Accuracy of training with frozen language model and co-finetuning

| Method | Robosuite | Metaworld |
|---|---|---|
| Co-finetune | 0.8489 | 0.8288 |
| Freeze T5 | 0.7625 | 0.7344 |

### C.2  Response Time to Different Feedback Types

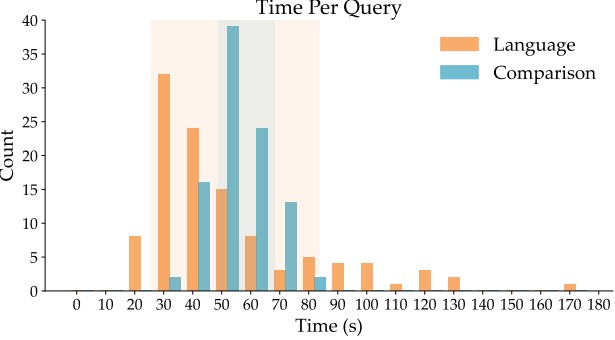

Figure 11: Time per query for both methods and averages, for 10 distinct users, indicated by color.

Figure 11 shows user-level differences of feedback length and time. The average of time of language-based reward learning is less than the comparison preference learning, illustrating that our method outperforms baseline in terms of the time-efficiency. However, the variance of language-based reward learning is larger than the baseline, which means users' thinking time varies a lot based on the trajectory shown.

# D    User Study Guidelines

**Introduction.** Welcome to our user study on human preference learning! In this study, you will interact with a robot and provide feedback based on your preferences. Please note that there will be no physical contact with the robot.

**Task Description.** The robot is situated in a kitchen setup and needs to pick up a spoon from the wall. However, there is a pan on the stove cooking something. For safety reasons, you should aim to have the robot successfully pick up the spoon while avoiding the pan. You can also adjust the robot's speed based on your preference.

**Pre-study Survey.** Before we begin, please complete a pre-study survey about your background and understanding of robotics and AI systems.

**Experiment 1 - Improve Trajectory.** In this experiment, the robot will first execute a suboptimal trajectory. Your task is to improve its behavior by providing comparative language feedback, such as "avoid the pan better." After each iteration, you will be asked if you are satisfied with the current trajectory. The experiment ends when you are satisfied with the trajectory or reach the maximum number of iterations. A post-study survey will be required after completing this experiment.

**Experiment 2 - Preference Learning.** In this experiment, you will be shown a series of queries and provide feedback on each one. Through this process, we will gradually learn your preferences and present you with the best trajectories based on the learned preferences. You will compare two approaches: language preference learning and pairwise preference learning.

1. Language Preference Learning
   - In each query, you will be shown one trajectory. You need to give comparative language feedback based on your preferences, such as "move faster," "avoid the pan better," or "be more adept at picking up the spoon." After every 5 queries, you will be shown the best trajectory based on the currently learned preferences and asked to rate it. You will complete a total of 20 queries.

2. Pairwise Preference Learning
   - In each query, you will be shown a pair of trajectories. You need to choose your preferred one. After every 5 queries, you will be shown the best trajectory based on the currently learned preferences and asked to rate it. You will complete a total of 20 queries.

After completing the experiment, you will be required to fill in a post-study survey.

# E    Original Language Feedback Utterances

All the original language feedback for each of the simulation environments are listed here. These sentences were then augmented with GPT-3.5.

**Robosuite:**

- Distance to the cube
  - Move farther from the cube.
  - Move further from the cube.
  - Move more distant from the cube.
  - Move less nearby from the cube.
  - Move nearer to the cube.
  - Move closer to the cube.
  - Move more nearby to the cube.
  - Move less distant to the cube.
- Distance to the bottle
  - Move further from the bottle.
  - Move farther from the bottle.

- Move more distant from the bottle.
- Move less nearby from the bottle.
- Move nearer to the bottle.
- Move closer to the bottle.
- Move more nearby to the bottle.
- Move less distant to the bottle.

- Height of the robot arm
  - Move taller.
  - Move at a greater height.
  - Move higher.
  - Move to a greater height.
  - Move lower.
  - Move at a lesser height.
  - Move shorter.
  - Move to a lower height.

- Speed of the robot arm
  - Move quicker.
  - Move swifter.
  - Move at a higher speed.
  - Move faster.
  - Move more quickly.
  - Move at a lower speed.
  - Move more moderate.
  - Move slower.
  - Move more sluggish.
  - Move more slowly.

- Proficiency at cube-lifting
  - Lift the cube better.
  - Lift the cube more successfully.
  - Lift the cube more effectively.
  - Lift the cube worse.
  - Lift the cube not as well.
  - Lift the cube less successfully.

**Meta-World:**

- Height of the robot arm
  - Move higher.
  - Move more up.
  - Move higher up from the table.
  - Increase the overall height of the trajectory.
  - Go higher up.
  - Move your gripper higher.
  - Don't go down, instead go up.
  - Stay higher and farther from the table.
  - Move your hand up as you perform the task.
  - Make sure to stay higher above the table, rather than lower.
  - Move lower.
  - Move more down.
  - Move lower to the table.

- Decrease the overall height of the trajectory.
- Go lower down.
- Move your gripper lower.
- Don't go up, instead go down.
- Stay lower and nearer to the table.
- Move your hand lower as you perform the task.
- Make sure to go lower to the table, rather than higher.

- Speed of the robot arm
  - Move faster.
  - Move at a quicker speed.
  - Increase the pace.
  - Press the button faster.
  - Increase your velocity.
  - Move your gripper faster.
  - Move to the button faster.
  - Don't go too slowly, instead go quickly.
  - Move your hand faster as you perform the task.
  - Make sure to go much faster, rather than slower.
  - Move slower.
  - Move at a more sluggish speed.
  - Decrease the pace.
  - Press the button slower.
  - Decrease your velocity.
  - Move your gripper slower.
  - Move to the button slower.
  - Don't go too fast, instead go more slowly.
  - Move your hand slower as you perform the task.
  - Make sure to go much slower, rather than faster.

- Distance to the button
  - Move farther from the button.
  - Increase distance from the button.
  - Stay farther from the button.
  - Give wider berth to the button.
  - Keep a larger distance from the button.
  - Keep your gripper farther away from the button.
  - Move your gripper away from the button.
  - Don't go towards the button, instead move away from it.
  - Move your hand farther from the button on the wall as you perform the task.
  - Make sure to go much farther from the button, rather than closer to the button.
  - Move closer to the button.
  - Decrease distance from the button.
  - Stay closer to the button.
  - Get closer to the button.
  - Keep a smaller distance to the button.
  - Keep your gripper closer to the button.
  - Move your gripper so that it is closer to the button.
  - Don't go away from the button, instead go towards it.
  - Move your hand closer to the button on the wall as you perform the task.
  - Make sure to go much closer to the button, rather than farther from the button.

