# OpenReview forum: "Trajectory Improvement and Reward Learning from Comparative Language Feedback"
_robot-learning.org/CoRL/2024/Conference — CoRL 2024_

### Official Review · Reviewer_1rvf · 2024-07-20
**A practical approach to learn improved trajectories from language feedback**

**Originality:** 3
**Technical Quality:** 3
**Clarity Of Presentation:** 4
**Potential Impact:** 3
**Recommendation:** 3
**Confidence:** 4

**Review:**

Strengths:
1. The paper takes a step towards answering how language feedback may be used to guide finding optimal trajectories much more efficiently than comparitive feedback. This setting is more practical and extremely relevant to real world robot learning.
2. A novel approach is proposed: Learn trajectory embeddings such that their difference is aligned with a language text that explains that difference. With a learned embedding, whenever an user gives a language feedback on the trajectory, many trajectory comparisons can be obtained by creating artificial trajectories and assuming the humans are giving the most informative feedback.
3. The algorithm is simple, based on supervised learning: It is tested for trajectory improvement using a pre decided dataset of trajectories and language feedback. Both synthetic and human experiments show that this method indeed achieved better performance compared to using comparitive feedbacks,


Weaknesses:
1. Obtaining supervision labels for training embedding model: Obtaining supervision labels for training the embedding model is a hard task. It is clear the using humans to label is not scalable and using VLM's to label can lead to hallucination. It is not discussed how the authors propose to solve this problem and consider a simple synthetic setting in their experiments.
2. Unable to handle ambiguity and lack of principled grounding: When a human gives a language feedback " move right" they may also want to "increase velocity" as another objective. By constructing fake labels we downweight the "increase velocity" and get stuck in suboptimal policies. The method thus lacks a principled grounding and it is not clear if optimal trajectories can be guaranteed.
3.  Policy optimization vs a toy setting: The authors consider an oversimplified setting of finding the best trajectory within a set of trajectories. While this setting makes the experiments easy it is an impractical one. In practice, it would be used in a policy optimization loop. How does the algorithm work in that setting remains an open question
4.  Situating in literature: The authors claim in the work they are the first to use language feedback for reward learning - this is untrue (See questions). Second, the authors perform a shallow literature review of the work of preference based learning, for example comparisons to various models that explore reward learning in PbRL from first principles are missing [1,2,3]

[1]: https://arxiv.org/abs/2202.03481
[2]: https://arxiv.org/abs/2310.13639
[3]: https://arxiv.org/abs/2206.02231

**Quality Of The Limitations Section:**

3

**Questions For Rebuttal:**

1. Overclaim of first method to use language feedback in reward learning: Some examples where language feedback has been used in prior works [1,2].


[1]: https://arxiv.org/abs/2310.17555
[2]: https://cdn.aaai.org/ojs/16749/16749-13-20243-1-2-20210518.pdf

**Robotics Focus:**

4

**Summary Of Paper:**

The paper propose a new approach to use the rich information contained in the language feedback to a trajectory and compare that to pairwise preferences.

**Summary Of Recommendation:**

Paper presents a new and simple solution but the experiments consider a toy setting making it far from real world world deployment.

---

### Official Review · Reviewer_e57u · 2024-07-20
**Refining trajectories by interpreting language feedback as difference between current and ideal trajectories. Lacks strong baselines.**

**Originality:** 4
**Technical Quality:** 3
**Clarity Of Presentation:** 3
**Potential Impact:** 3
**Recommendation:** 3
**Confidence:** 4

**Review:**

**Strengths**

- The proposed idea of interpreting language as a comparison or difference between the desired and current trajectory, and implementing it through a shared latent space is novel and useful
- The dual interpretation of human feedback, that the improved trajectory is more preferable than both the current trajectory, as well as the trajectories resulting from alternative language feedbacks is innovative, and the ablation results empirically show how each of those two kinds of information help the final model performance
- The attached video clearly outlines major advancements proposed in the work

**Concerns**

- Several key details of the evaluation are unclear.
     - “Lists of language feedback were created for hand-crafted features  (see Appendix A)” → How are the language corrections created? Appendix A, is not present in the submission. Also, what do the authors mean by “[the language feedback] were augmented with GPT 3.5”? How are the language corrections matched with the trajectory rollouts?
     - For reward learning, what are the features ($\theta$)? How are those generated and what is the dimensionality of \theta? Is it correct that a randomly sampled vector $w$ represents a simulated human? And if so, how many simulated humans is the evaluation done on?
     - In simulated results, can the authors explain why the cross entropy does not drop past 0.4, especially since the function simulating human preferences matches the loss function being learned. Also, why is cross-entropy used to evaluate reward learning accuracy rather than something more direct, such as % error in predicted reward, compared to the true reward, since the true reward is available?
     - For the user study, how were the participants instructed to determine their preferences, especially since it is difficult to evoke preferences in short horizon tasks in a lab setting. Can the authors show what kind of preferences are expressed, and how diverse the preferences and the set of sampled trajectories were?
- No other work utilizing language feedback is used as a baseline. Several methods that utilize natural language as feedback to improve trajectories are mentioned in related work (such as [13,12]). While these methods utilize feedback in different ways, since they fit the same problem formulation of finding a better trajectory given language feedback, why is none of them used as a baseline?
- The problem of improving a robot trajectory and learning a human preference are fundamentally the same, in that some human correction needs to be used to find a better trajectory, and accordingly, both iterative trajectory improvement (4.2.1), and reward learning (4.2.2) serve to ultimately find an optimal trajectory given human corrections. Then why are both methods highlighted, since iterative trajectory improvement clearly performs worse than the latter?
- The ability to find the optimal trajectory in a real setting should be the core empirical result. Figure 10 alludes to this but is not referenced anywhere in the text. Can the authors elaborate on it, especially why the two methods end up with very similar performance at 20 iterations?

**Other Clarifications and Suggestions**

- The related work section can be better structured, and highlight works on language-based robot manipulation better. The ‘Learning from Human feedback’ section talks about robots and seems redundant with ‘Robot Learning with Human Feedback’. It would also help to elaborate on the works that use natural language to improve robot policies, and how exactly the proposed method is positioned in context of those methods. Related works on using language corrections to guide robot control such as the following, amongst others, could be highlighted.
[1] Lynch, Corey, et al. "Interactive language: Talking to robots in real time." IEEE Robotics and Automation Letters (2023).
[2] Cui, Yuchen, et al. "No, to the right: Online language corrections for robotic manipulation via shared autonomy." Proceedings of the 2023 ACM/IEEE International Conference on Human-Robot Interaction. 2023.
- In Section 3, it is unclear how language feedback relates to the reward framework discussed before it.
- In Section 4.1. eq(3), how does optimizing the output of the network to be normalized prevent overfitting? Traditionally, regularization constraints are applied to network parameters and not network outputs.
- The authors emphasize how comparative feedback contains 1-bit of information. I wonder if it is possible to quantify statistical information gain from a language feedback, especially in the evaluation setting which has a fixed number of trajectories.
- In Section 4.2.2, the terminology of ‘implicit’ referring to direct comparison against current trajectory, and ‘explicit’ referring to comparison against trajectories resulting from other possible language corrections, seems upside-down intuitively, since the former is a more ‘explicit’ meaning of the correction.
- In Section 5.1, eq(12), what is the softmax being computed across? And how does the function $l$ generate language feedback?

**Edit after rebuttal:** The authors answered questions and clarified the necessary details.

**Quality Of The Limitations Section:**

3

**Questions For Rebuttal:**

See questions in review, particularly those regarding:
- Missing details in the evaluation sections
- Using language-based baselines
- Explaining results in Figure 10 and justifying the narrow margins in performance improvement
- The difference between iterative trajectory improvement, and reward learning, and why both aspects are important to include

**Robotics Focus:**

4

**Summary Of Paper:**

The authors propose a method to leverage language feedback to improve robot trajectories to better satisfy human preferences. Instead of seeking a binary comparative feedback, as popularized by RLHF-style methods, this work seeks to obtain natural language feedback, which hypothetically might contain more information than a binary comparative label. The authors propose to learn a mapping in latent space between correction in language space and that in trajectory space. They then use this latent space mapping to learn a reward function, interpreting language correction as a preference between the two corresponding points in the trajectory space. To learn the reward function they use the Bradley-Terry model, commonly used for comparative feedback. They evaluate their model on simulated data on two tasks, and through a user study.

**Summary Of Recommendation:**

The paper proposes a novel method to incorporate language feedback in iteratively improving trajectories to better fit a human’s preferences. The work proposes a viable formulation and promising method, but evaluations could include language-based baselines and several details regarding the method and evaluation are unclear.

---

### Official Review · Reviewer_KGNz · 2024-07-30
**Review of Trajectory Improvement and Reward Learning from Comparative Language Feedback**

**Originality:** 3
**Technical Quality:** 4
**Clarity Of Presentation:** 4
**Potential Impact:** 3
**Recommendation:** 3
**Confidence:** 4

**Review:**

Overall I think this paper was clearly written and addresses a novel yet challenging problem of understanding human preferences from language in robotics setting. I thank the authors for their clear presentation of their work. In my opinion this paper has the following strengths

Most important strength of this paper was the novelty and the difficulty of the problem space in my opinion. As humans we communicate our preferences to each other by language and I do agree that this would be a much preferred way of communicating our preferences to robots as well. On the other hand learning humans reward functions from these language utterances is a challenging problem. I think authors present an interesting attempt at solving this problem by pairing the trajectory embedding differences with the language embedding to learn an aligned latent space representation. Another strength of this paper showing usability and preference of actual users by running a user study. The user study is important to back the claims of comparative language being more natural to the users. Lastly I think overall authors did a good job on clarity and quality when it comes to writing their work.

In terms of weaknesses I think one major weakness in my point of view was the authors did not report significance of the difference in their user study results. I would like to see statical test results mentioned and explained. Also I think 10 participants might be too little. I think having a bigger and more open user study would have benefitted this work. Another weakness in my view was the use of very simple tasks in the simulated settings I think authors could have shown the method in more complicated domains where language utterance can also significantly vary.

**Quality Of The Limitations Section:**

3

**Questions For Rebuttal:**

My main question for the authors are the statistical test results on the user study survey responses. I think that should be reported even if the subject size is quite small. Another point to address would be how would this method work in settings where many language utterances can be mapped to a single trajectory update? How well can this system generalize in terms of the language utterances it can understand? What is the benefit of co-finetunining versus using a frozen language model for getting the embeddings?

**Robotics Focus:**

4

**Summary Of Paper:**

This paper aims to propose a novel way of doing preference based learning by utilizing comparative language as the feedback rather than traditional comparison based approaches. For this purpose authors created a model that leverages learning the latent state representations that align language utterance and changes in robot trajectory. Then this model is used to both iteratively update the robot trajectories as well as learning humans reward model. Authors backed their proposed approach with simulated and real-world experiments.

**Summary Of Recommendation:**

I recommend this paper to be accepted given the novelty and importance of the subject as well as the good execution and testing done by the authors.

---

### Author Rebuttal · Authors · 2024-08-10

We want to thank the reviewers and the area chair for the valuable feedback. We are happy that all reviewers acknowledge that the problem we are trying to solve is challenging and important, and our approach is novel. We also appreciate the praise for our real-world user study for evaluation. Moreover, reviewer KGNz found our idea of utilizing comparative language feedback to communicate to the robot more natural than existing approaches, reviewer e57U stated the attached video clearly outlines the advances of the paper, and reviewer 1rvf said our approach is practical and relevant to the real-world robot learning.

We have updated the manuscript based on the reviewers' comments. The new version is attached in this comment.

We respond to the individual reviewers' comments with official comments under their reviews.

---

### Decision · Program_Chairs · 2024-09-04

**Decision:**

Accept

**Comment:**

**Paper summary**

This paper presents a method for using language feedback to indicate and learn from preferences between trajectories. The results indicate that language feedback provides more informative data than preferences alone, leading to more efficient and user-preferred learning performance.

**Review summary**


Summary of strengths:
+ Learning effectively from language feedback is a highly challenging, yet important, problem.
+ The proposed method is novel in that it aims to learn a latent space mapping from language feedback to trajectory modifications.
+ The evaluation involves a user study with non-expert users.

Summary of weaknesses:
- The paper is missing important comparisons to prior work on learning from language. Of the papers that are cited, none are used as baselines in the evaluation. **[Edit: the revised paper includes several new citations to relevant SOTA work. However, none are evaluated as baselines.]**
- The proposed method must be trained on labeled data, yet it is unclear how this would be obtained in a scaleable way. **[Edit: the rebuttal proposes two ways to improve the scaleability of data collection.]**
- The paper does not discuss how the proposed method would be integrated into a policy optimization problem (which would seemingly be the goal of learning from language feedback). **[Edit: the rebuttal argues that its sampling-based method is similar to other SOTA approaches for generating and evaluating trajectories.]**

**Response to rebuttal**

Overall, the reviews are positive and remain so after the rebuttal. The rebuttal addressed the reviewers' requests for clarifications and added citations.

However, the rebuttal does not resolve the request for a baseline method that uses language feedback. The authors claim that existing approaches are not suitable as baselines due to the lack of published code, but this is not a sufficient reason to exclude otherwise-relevant SOTA baselines.


**Recommendations for improvement**

* Add baselines to contextualize the results with respect to SOTA methods for learning from language.